# Does an improved HDI trigger tourism outflows for China? New evidence from the ARDL cointegration approach

Ahsan Akbar[1,2], Farrukh Nawaz[3], Xie Hui[4]*, Irfan Ullah[5], Minhas Akbar[2,6], Veronika Zidova[2], Asokan Vasudevan[6]

1 International Business School, Guangzhou City University of Technology, Guangzhou, China, 2 Department of Recreology and Tourism, Faculty of Informatics and Management, University of Hradec Kralove, Hradec Kralove, Czechia, 3 Faculty of Business Studies, Arab Open University, Riyadh, Saudi Arabia, 4 School of Jewelry, Guangzhou City University of Technology, Guangzhou, China, 5 Reading Academy, Nanjing University of Information Science and Technology, Nanjing, People's Republic of China, 6 Faculty of Business and Communications, INTI International University, Nilai, Negeri Sembilan, Malaysia

* xiehui@gcu.edu

## Abstract

Extant studies have predominantly focused on understanding the effects of inbound tourism on economic growth. At the same time, it ignores the key factors that promote outbound tourism in a country. Outbound tourism not only plays a crucial role in the sustainable development of the host country but also helps foster an understanding of cross-cultural similarities and differences, promoting goodwill towards the home country. It also provides an opportunity to experience the cuisine, weather, and working habits of different geographical locations. Therefore, this study aims to investigate the relationship between the Human Development Index and tourism outflows in the context of China. We use data from the period 1995–2020 to estimate the relationship between the Human Development Index (HDI) and tourism outflows. The short-run and long-run Autoregressive Distributed Lag (ARDL) results indicate that improving China's Human Development Index (HDI) has a positive impact on tourism outflows. More precisely, a 1 unit increase in the HDI index will increase tourism outflow by 5.25%. Furthermore, the results of individual HDI show that life expectancy, per capita income, and education considerably promote China's tourism outflows. Policy prescriptions are outlined regarding the spillover effects of China's human development on the tourism-led economic growth of recipient countries.

## 1. Introduction

In this era of globalization, the world tourism industry has grown exponentially, contributing significantly to the economic growth of various countries. The tourism industry worldwide is flourishing and booming because of technological developments [1]. Technological developments have made traveling more frequent and fascinating

**Data availability statement:** All relevant data are within the paper and its Supporting Information files.

**Funding:** This research is supported by the specific project 6/2024 grant "Determinants of Cognitive Processes Impacting the Work Performance" granted by the University of Hradec Kralove, Czech Republic and thanks to help of student Ing. Frantisek Hasek; a Long-term development plan of the University of Hradec Kralove.

**Competing interests:** The authors have declared that no competing interests exist.

by helping tourists prepare and execute the most successful travel plans. For the sustainable development of an economy, the planning phase and, accordingly, the development intervention play a significant role [2]. This planning and development phase is fruitful when it reaps various objectives for the economy concerned, specifically by reducing poverty, enhancing economic growth, and fostering human development [3,4].

According to the United Nations World Tourism Organization [5], the international tourism industry has expanded rapidly in recent years. Tourism contributes to the welfare of individuals and society through the active participation of the local communities, empowerment, and efficient resource management [5]. These benefits are helping societies to achieve human development, defined as the ability to promote and enhance the mobility of individuals [3,6]. The Human Development Index [7] was first proposed by [8]. It has been widely used in applied economics literature since then. HDI comprises three main indicators of human development: life expectancy at birth, literacy rate, and per capita income of the residents of a country. Hence, HDI presents a holistic picture of humans' socio-economic development and well-being rather than a given economy's narrow definition, such as GDP growth [9].

China has experienced remarkable economic growth since adopting its policy of opening up in 1978, under the leadership of visionary President Deng Xiaoping at the time. Deng transformed the Chinese economy from a socialist to a mixed economy, combining elements of capitalism and socialism, to achieve economic progress and prosperity [10]. One of his essential and transformative sayings is worth mentioning: "It does not matter whether the cat is black or white as long as it catches the mice." [11]. Due to his economic vision and the transformation of China, the living standards of the Chinese people have markedly improved over the last 30–40 years. Likewise, the personal disposable income of Chinese people has increased due to the country's exponential GDP growth. This, in turn, has increased the affordability of outbound tourism for people [12,13].

China's tourism outflow dates back to 1984, when visiting relatives living in Hong Kong and Macau was a common practice. These outbound visits were sanctioned at the start. However, in 1990, the China National Tourism Administration (CNTA) announced "Provisional measures concerning the administration of arranging Chinese citizens to three Southeast Asian countries namely, Thailand, Singapore and Malaysia". In continuation, in 1997, the "Provisional measures concerning the administration of outbound travel of Chinese citizens at their own expense" were promulgated [14]. These initiatives led to the establishment of the Approved Destination Scheme (ADS) and a surge in Chinese outbound tourism. As per the 2015 report by the United Nations' World Tourism Organization (UNWTO) [5] China has become the leading player in the international flow of tourism and has surpassed many advanced countries, such as Germany and the United States [15,16].

Before delving into further details regarding China's tourism outflow, it is pertinent to mention that China's tourism inflows have also experienced exponential growth, promoting economic growth through increased foreign exchange earnings [17]. Over the last five years, the outflow of tourism in China has grown significantly, increasing

by 14.1% compared to the 11.6% inflow of tourism [15]. Based on this phenomenal increase in tourism in China, the researchers have shown considerable interest in assessing the underlying factors of tourism outflows from China [18]. However, the extant literature has predominantly focused on understanding the effects of tourism outflows on the recipient country's economic growth, while ignoring the role of human development and improvement in quality of life [19].

Prior studies have mostly examined the relationship between economic growth and outflow of tourism, and three possibilities can be found in the literature. A unidirectional causality between tourism and growth [20–28], unidirectional causality between economic growth and tourism [29,30], or bidirectional causality between tourism and growth [31–33]. Other studies found that depreciation in the value of a country's currency boosts the balance of tourism trade [34]; children from the host country have a negative attitude towards tourists [35]. Solo female tourists are more susceptible to fear and anxiety [36]. Moreover, an increase in inbound tourism significantly increases CO2 emissions in Italy [37]. Based on the above studies, it is evident that tourism is widely viewed and examined in the context of economic growth, whereas HDI fails to attract the attention of researchers [38]. Even fewer studies have explored the impact of tourism on HDI and economic growth [39–41]. Others examined the influence of HDI on carbon emissions and healthcare spending [42] and energy consumption on healthcare spending [43]. However, it remains unclear how improvements in HDI spur the demand for outbound tourism from a country.

Therefore, the present study is novel in the context as it is perhaps the first attempt to explore the impact of HDI on tourism outflows through an in-depth examination of the world's largest contributor to tourism outflows, i.e., China. The present research contributes to the tourism outflow domain by empirically testing the impact of improved HDI on China's outbound tourism through short-run and long-run ARDL estimations for 1995–2018. Furthermore, the life expectancy at birth, literacy rate, and per capita income components of China's HDI are individually tested, along with tourism outflow, to ensure the robustness of our empirical outcomes.

The remainder of this article is organized as follows: Section 2 presents a synthesis of the literature review. Section 3 explicates the trend analysis of tourist outflows and HDI components. Section 4 expounds the methodology, which uses an ARDL approach. Section 5 presents the results and the discussion. The final section concludes the article with the implications of this research.

## 2. Literature review

In the current era, outbound tourism is recognized as a vital contributor to a country's economic growth. Following the 1978 economic reforms and China's open-door policy, tourism has been considered as a crucial source of economic development. For bilateral tourism agreements between China and overseas destinations, Chinese outbound tourism has been managed and regulated by the Approved Destination Status (ADS) system [44].

Chinese outbound tourism increased in number from 1994, with an average tourism outbound flow of 6,106.600 person to an all-time high of 161,993.600 person in 2018 [45]. According to [46]Chinese outbound tourism has an impact on Western research, and shortly, it will contribute to the generation of new theories in the domain of international tourism.

In the literature, scholars have explored the relevance of tourism and macroeconomic variables using various methods and econometric techniques, including ARIMA models, Cointegration analysis, MGARCH, and OLS techniques. The impact of external political, economic, and environmental crises and their effects on Chinese outbound tourism was examined by [47]. The study employed time-series data and in-depth interviews to examine the negative impacts of the crisis on tourism and the underlying factors contributing to the industry's growth. They found various negative impacts on the tourism sector caused by political and economic crises. The expectation of future wealth and its impact on outbound tourism in Korea was investigated from 1989 to 2009 [48]. The findings posit that the wealth of households positively impacts outbound tourism. Dai, Jiang [17] explored China's outbound tourism policies, stages, and choices, and found that outbound tourism has a strong impact on the future orientation of China's tourism policy.

Likewise, international trade activity can also influence the inflows of tourists in a country. [49] noted that the Germans' tourism in Spain was positively linked to the wine import into Germany. Additionally, the findings of the long memory regression model suggest that this effect persists for two to nine months. Though, [50] noticed that the positive impact of tourism arrivals on trade withers away in the long run. However, a long-run equilibrium was observed between tourism flows, imports, and exports in the context of South Asian countries. These findings suggest that exports and imports increase a country's recognition and image in foreign markets. Similarly higher imports make it easier for foreign tourists to find their local products in the host country [51].

[52] The panel cointegration technique was used to reveal that tourism and international trade flows are strongly related in the context of Turkey and Silk Road countries.

Lim [53] found that most studies used tourist arrivals, departures, and expenditure receipts as dependent variables, while the explanatory variables included income, relative tourism prices, and transportation costs. Song, Romilly [54] introduced the tourism destination preference index, which examines the social, cultural, and psychological influences on tourists' decisions regarding their foreign holiday destinations, and found long-running relationships for each of the sampled destinations. Cortés-Jiménez, Durbarry [55] used a monthly Dynamic EC-LAIDS model to investigate the outbound demand for Italian tourism in terms of responsiveness to relative prices, exchange rates, and expenditure changes for four European destinations. Findings reveal that short-term elasticities are crucial for cross-price and expenditure policies. Moreover, the EC-LAIDS model outperforms the long-run model in forecasting accuracy. Seo, Park [56], the relationship between Korean outbound tourism demand for Jeju Island and three international destinations was explored using Multivariate Generalized Autoregressive Conditional Heteroskedasticity (MGARCH) and Vector Error Correction (VEC) analyses. They found that, in certain time horizons, Jeju Island serves as a substitute tourist destination for three Asian countries. The results of the VEC model further indicate that the industrial production index and real exchange rates have a substantial impact on the conditional correlations for these destinations.

Seetaram [57] explored the demand elasticity of international outbound tourism for Australian tourists. He developed a dynamic model for international travel from Australia, using 47 destinations from 1991 to 2008. This study revealed that immigration laws play a significant role in influencing international tourist departures from Australia, both in the short and long term. Lin, Liu [58] examined the factors affecting the demand for outbound tourism from mainland China to international destinations. Their findings assert that income levels and the cost of stay at tourism destinations were the main factors impacting China's outbound tourism. Likewise, [59] found that education, GDP, internet penetration, vacations, and per capita income are influential factors in China's outbound tourism to North Korea. and Lee [60] explored the factors affecting the choice of outbound tourism destinations for Hong Kong citizens. They found that trip expenditure, household income of tourists, stress, and the desire to escape routine life significantly affect tourists' destination choices. The above studies highlighted that outbound tourism is significantly affected by the tourists' income, wealth, and health.

Recent empirical studies have examined the relationship between the Human Development Index (HDI) and tourism outflows, underscoring the crucial role of development and prosperity as a multifaceted factor in international tourism promotion. Zhao et al. [61] observed that in countries along the Belt and Road, the improvement of HDI in the domains of income and education considerably enhances outbound tourism, since these factors directly influence individuals' tourism capacity and intention. Likewise, Tan et al. [62] found a long-term positive association between HDI and outbound tourism in Malaysia, accentuating that the development of education and health improved the conditions and motivation of outbound tourists from Malaysia. Wibowo et al. [63] upheld the notion that tourism-led economic growth and human development aspects are mutually reinforcing, forming a bi-directional linkage in their research on ASEAN countries. Further expanding on this viewpoint, [64] revealed a nonlinear U-shaped relationship between HDI and outbound tourism, indicating that the impact of HDI on outbound tourism is significantly enhanced after reaching a certain level of development. These findings highlight the significant impact of HDI and its various components on tourism outflows. Hence, it is suggested that improved human development conditions not only

facilitate but also stimulate international mobility from a country. Against the backdrop of a remarkable upsurge in the human development indicators [61] and a considerable growth in the outbound tourism from China in recent years [46] the present study adds value to the existing literature by examining the influence of HDI as well as its components of Education, Income, and life expectancy at birth on the growth in outbound tourism from the world's second-largest economy.

## 3. Tourist outflow, HDI components trend analysis

### 3.1 Tourism outflow and income

According to the China Tourism Academy, in 2018, China remained the world's biggest market in outbound tourism, with nearly 150 million outbound visits made by Chinese travelers. Despite several factors behind this trend, income is a significant factor that stimulates tourist outflow activities. China's booming outbound tourism market owes much to the country's rapid economic growth, that led to a rise in the standard of living and disposable income [65]. Fig 1 presents a graphical representation of the pattern of tourist outflows and per capita income in China from 1995 to 2017, sourced from the World Bank's development indicators. In 1995, the tourist outflow from China was a mere 4.5 million; this figure has continued to increase until 2017, when it reached 130 million. It shows a rapid increase in tourism activities from 2008 until 2013; a moderate increase has been witnessed thereafter. Income also shows a steady growth from 1995 to 2018. The co-movement of both variables conjectures an association between income and tourism outflows in China. However, tourist outflows increase with relatively higher elasticity than income, indicating that other factors may stimulate the tourism outflows besides income.

### 3.2 Tourist outflows and life expectancy

Life expectancy reveals the health and well-being aspect of the tourist. It has been observed that public spending on environmental protection helps increase life expectancy in China [66]. Moreover, healthcare spending also reduces the death rate [67]. Health quality and longevity can lead to an active lifestyle, thus promoting the urge to engage in tourism activities. Fig 2 presents the life expectancy and tourist outflow from 1995 to 2018. The life expectancy data reveals a linear trend with no significant deviation. However, tourism outflows also show an increasing but volatile behavior. Nevertheless, there appears to be some association between the two series during the study period.

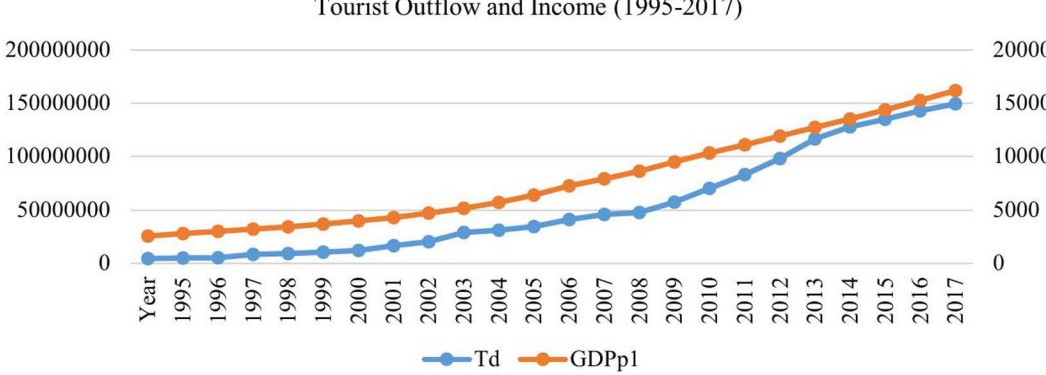

**Fig 1. Tourist outflow and income level.**

## 4. Methodology

### 4.1 Econometric model

The empirical model of this study can be mathematically expressed as follows:

$$TD_t = \beta_0 + \beta_1 HDI_t + \beta_2 Ex_t + \beta_3 GDP_t + u_t \tag{1}$$

Where

TD = Tourist outflow

HDI = Human Development Index

Ex = Exchange rate

GDP = Gross Domestic Product

$u$ = is the error term

In our proposed model, the tourism outflow is taken as the dependent variable, while HDI is our main explanatory variable. Moreover, we also employ control variables such as GDP and exchange rate, as they can significantly influence the tourism outflow. GDP is expected to have a positive association with TD. The exchange rate is expected to have a negative association with TD, which means that, with exchange rate depreciation, the real value of the local currency depreciates compared to the foreign currency, implying that tourism activities become costlier. HDI is expected to positively impact the tourism outflow, as increased education, life expectancy, and disposable income will make people more capable, intellectually, physically, and financially, to undertake foreign trips.

### 4.2 Empirical methods

We employ the Autoregressive Distributed Lag (ARDL) model as a baseline analysis. The ARDL approach can estimate both long-term and short-term relationships between the variables. The co-integration method was initially proposed by Engle and Granger [68] is based on the assumption that all the variables shall hold a first order of integration (I(1)). It implies that all the variables must be integrated at first difference. However, the EG method doesn't provide information on the number of co-integrating vectors. Johansen [69] relaxed the assumption of order 1 integration and suggested a co-integration that provides information on the number of co-integrating vectors. Johansen test uses Maximum Likelihood Estimates (MLE) to examine the number of connections among the study variables.

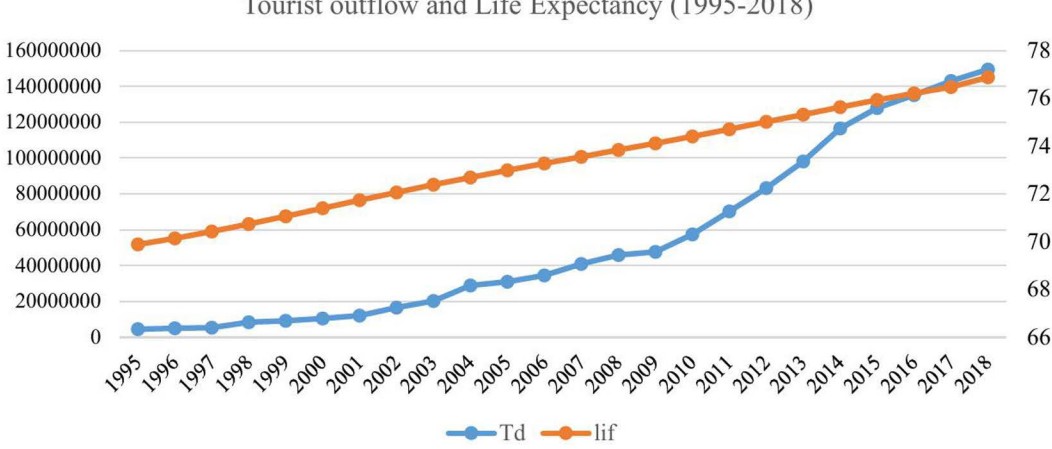

**Fig 2. Tourist outflow and life expectancy.**

Pesaran, Shin [70] proposed the ARDL method to identify the co-integration of variables even if the variables do not hold the same order of integration. Besides, this method improves the reliability of small sample sizes and provides an accurate estimate of the long-term estimates of the underlying model. Similar technique was used by [71] while examining the impact of economic growth, energy consumption, urbanization, population and tourism on $CO_2$ emissions in the context of Malaysia. Likewise, similar technique was adopted by recent studies examining the association between tourism and other macro-economic variables in the context of time-series data [72,73]. We apply the ARDL bound test for equation 1 as follows:

$$\Delta TD_t = \alpha_0 + \sum_{i=1}^{n-1} \alpha_{1i}\Delta TD_{t-1} + \sum_{i=1}^{n-1} \alpha_{2i}\Delta GDP_{t-1} + \sum_{i=1}^{n-1} \alpha_{3i}\Delta Ex_{t-1}$$
$$+ \sum_{i=1}^{n-1} \alpha_{4i}\Delta HDI_{t-1} + \beta_1 TD_{t-1} + \beta_2 GDP_{t-1} + \beta_3 Ex_{t-1} + \beta_4 HDI_{t-1} + ut$$

(2)

The above equation incorporates the difference operator $\Delta$ for each respective variable, and $\alpha0$ is the deterministic drift parameter. To test the existence of cointegration between TD, GDP, Ex and HDI we employ the following null hypothesis: $H_0$): $\beta1=\beta2=\cdots=\beta6=0\beta1=\beta2=\cdots=\beta6=0$ (no cointegration), against the alternate hypothesis $H_1$): At least one of the $\beta i$ is non-zero (cointegration exists) by using F-statistics developed by Pesaran, Shin [70] and further modified by Narayan [74] for small samples. If our estimations reject the null hypothesis of no cointegration in equation 2, we can proceed with the error correction model as follows:

$$\Delta TD_t = \alpha_0 + \sum_{i=1}^{n-1} \alpha_{1i}\Delta TD_{t-1} + \sum_{i=1}^{n-1} \alpha_{2i}\Delta GDP_{t-1}$$
$$+ \sum_{i=1}^{n-1} \alpha_{3i}\Delta Ex_{t-1} + \sum_{i=1}^{n-1} \alpha_{4i}\Delta HDI_{t-1} + \gamma ECT_{t-1} + vt$$

(3)

Where $\gamma$ represents the speed of adjustment, while ECT residuals were obtained from the estimated model 2. The $\gamma$ is assumed to have a negative expected sign, allowing the model to return to long-term equilibrium quickly.

We use the same ARDL model for the robustness test, taking into account the individual components of HDI, such as literacy/expected years of schooling, life expectancy, and income, for the Chinese people. The data for all variables were obtained from the World Bank database and the United Nations HDI tables. Table 1 presents the description and sources of the study variables.

**Table 1. Variable descriptions.**

| Variables | Description | Source |
|---|---|---|
| TD | International tourism, number of departures | World Development Indicators |
| GDP | Annual GDP growth | World Development Indicators |
| EX | Real effective exchange rate index (2010 = 100) | World Development Indicators |
| HDI | Human Development Index | United Nations development program |
| Income | GNI per capita | United Nations development program |
| Lif | Life expectancy at birth, total (years) | United Nations development program |
| School | Expected years of Schooling | United Nations development program |

## 5. Results and discussion

This section provides a detailed discussion of empirical results. As a prerequisite, Table 2 provides the outcome of the ADF unit root test. The ADF unit root test indicates that all variables are not stationary in their levels, but become stationary at first difference, as evidenced by GDP and the exchange rate.

### 5.1 Baseline model results

The second step in the ARDL procedure is to test the existence of cointegration. The F-statistics are obtained by imposing restrictions on the coefficients of the baseline model. Table 3 contains the results of the optimal lag selection criteria. Table 4 exhibits the outcome of the bound test, which shows that F-values exceed the upper bound at both the 5 per cent and 10 per cent levels of significance, implying a long-term relationship between TD, GDP, Ex, and HDI.

**Table 2. ADF Unit Root Test.**

| Variables | ADF at level | ADF at 1$^{st}$ difference |
|---|---|---|
| TD | −2.991 (−1.95) | ------ |
| GDP | −0.759 (−2.99) | −4.370 (−1.95) |
| EX | 0.979 (−1.95) | −3.540 (−1.95) |
| HDI | −5.056 (−3.00) | ------ |
| Income | 14.835 (−1.95) | |
| Lif | 3.407 (−1.95) | ------- |
| School | −7.035 (−2.99) | ------- |

Critical value at 5% in the parenthesis.

**Table 3. Optimal lag selection criteria.**

| Variable | Lag | AIC | HQIC | SBIC |
|---|---|---|---|---|
| TD | 2 | 37.1708* | 37.2058* | 37.3196* |
| GDP | 1 | .533241* | .558073* | .631979* |
| EX | 1 | −3.55994* | −3.53511 | −3.46121 |
| HDI | 3 | −9.84798* | −9.79589* | −9.65164 |
| Lif | 2 | −1.55155* | −1.51248* | −1.40429 |
| Income | 3 | 13.9394* | 13.9915* | 14.1357* |
| School | 2 | −8.30195* | −8.2669* | −8.15317* |

**Table 4. Bound Test.**

| F-statistic | | 8.289 |
|---|---|---|
| Test Statistic | | −3.153 |
| Critical Value Bounds | | |
| Significance | Lower Bound | Upper Bound |
| 5% | 3.23 | 4.35 |
| 1% | 4.29 | 5.61 |

Note: Null Hypothesis: No long-run relationships exist.

## 5.2 Error Correction Model (ECM)

The third step in the ARDL method is to test the short-term dynamics in the model, which lead to the long-term equilibrium. The model below indicates the short-term coefficients and error correction terms. The variables are different, and short-term dynamics are presented in ECT (−1). The lag values of the tourist outflows (ΔTA (−1)) variables are positively and significantly linked with the dependent variable, implying that each previous period is a significant determinant of the subsequent year's tourist outflows. GDP and exchange rate are also positive and significant determinants of tourist outflows, indicating that tourist outflows significantly increase with increased GDP and exchange rates.

Notwithstanding, HDI has a significant impact on tourist outflows, implying that China's tourist outflows increase substantially with an increase in HDI. This is consistent with our expectation, as better education, health, and income drive more people to engage in excursion activities abroad. The error correction term is negative and significant, implying the existence of short-term dynamics that will restore the system to its long-term equilibrium shortly.

## 5.3 Long-term results

Table 5 shows the long-term results of HDI, GDP, and Ex. The empirical outcomes reveal that GDP has a significantly positive coefficient with tourist outflows; this implies that, with overall economic growth, tourist outflows tend to increase by 3.73% against a 1-unit increase in GDP. The exchange rate also exhibits a positive and significant association with TD. HDI is a positive and significant determinant of tourist outflows, reflecting that, with increased education, life expectancy, and income, Chinese people tend to participate more actively in overseas tourism activities. A 1% increase in HDI leads to a 5.25% rise in tourist outflow.

## 5.4 Robustness testing

We perform a component-wise analysis of HDI indicators to verify the baseline results, as HDI comprises life expectancy, education, and income. We use the same ARDL model and incorporate separate HDI components. Following similar empirical steps to ARDL, we first test the existence of a long-term relationship. Table 6 presents the results of a bound

**Table 5. Long-run coefficients for HDI.**

| Variable | Coefficient | Std. Error | t-Statistic | Prob. |
|---|---|---|---|---|
| HDI | 5.255399 | 1.716302 | 3.06 | 0.028 |
| EX | 9.039857 | 2.541221 | 3.56 | 0.016 |
| GDP | 3.732518 | 1.130126 | 3.30 | 0.021 |

**Table 6. Bound Test.**

| Model | Null Hypothesis: No long-run relationships exist | | Critical Value Bounds | | |
|---|---|---|---|---|---|
| | Test Statistic | Value | Significance | Lower Bound | Upper Bound |
| $TD_t = \beta_0 + \beta_1 Income_t + \beta_2 Ex_t + \beta_3 GDP_t$ | F-statistic | 7.017 | 5% | 3.23 | 4.35 |
| | | | 1% | 4.29 | 5.61 |
| $TD_t = \beta_0 + \beta_1 life_t + \beta_2 Ex_t + \beta_3 GDP_t$ | F-statistic | 6.173 | 5% | 3.23 | 4.35 |
| | | | 1% | 4.29 | 5.61 |
| $TD_t = \beta_0 + \beta_1 school_t + \beta_2 Ex_t + \beta_3 GDP_t$ | F-statistic | 6.997 | 5% | 5% | 3.23 |
| | | | 1% | 1% | 4.29 |

test in three separate models, incorporating income, life expectancy, and expected years at school, respectively. The null hypothesis, that no long-term relationship exists, has been rejected in all three cases, implying that our robust testing results are consistent with the findings of the baseline ARDL model.

**5.4.1 Error correction model.** The Error Correction model is used to exhibit the short-term deviation in the model. Table 7 presents the results of the ECM; we estimate three different models for each HDI component. The Error Correction term is negative and significant in all cases, indicating that the models' short-term dynamics exist and will achieve long-term equilibrium in the near future. Interestingly, the life expectancy aspect of HDI has a negative and significant relationship with tourist outflows in the short term; similarly, per capita income is positively and significantly linked with tourist outflows in the short term. Education, however, does not influence the tourist outflow in the short term.

**5.4.2 Long-run association results.** Table 8 demonstrates the long-term association between HDI components and outbound tourism from China. Life expectancy has a positive and significant association with tourist outflows, which implies that, in the long term, longevity tends to increase tourist outflows. Furthermore, this argument also holds for health

**Table 7. ARDL Error correction model.**

| Variable | TD | TD | TD |
|---|---|---|---|
| ΔLIF | −2.34 ** (0.052) | | |
| ΔIncome | | 2.65** (0.045) | |
| Δschool | | | −1.93 (0.149) |
| ΔTD | 2.48 ** (0.043) | 2.86** (0.035) | 2.86* (0.064) |
| ΔGDP (−1) | 1.66 (0.140) | 1.55 (0.181) | −2.57* (0.082) |
| ΔEX (−1) | −0.28 (0.789) | −2.25* (0.065) | −4.11** (0.026) |
| ECT(t-1) | −10.39 *** (0.00) | −4.61*** (0.006) | −4.86** (0.017) |

t-statistics in the parenthesis, ***, **, * denotes $p < 0.001$, $p < 0.05$ and $p < 0.1$, respectively.

**Table 8. Long run Estimations.**

| Dependent Variable | TD | TD | TD |
|---|---|---|---|
| Variable | Coefficient | Coefficient | Coefficient |
| Income | 1.547 *** (13.62) | | |
| Life | | 33.053*** (10.66) | |
| School | | | 3.677* (3.00) |
| Ex | 9.039** (3.56) | 1.615 (1.34) | 7.671* (2.606) |
| GDP | 3.732** (3.30) | 0.820 (1.61) | 3.359* (1.291) |

t-statistics in the parenthesis, ***, **, * denotes $p < 0.001$, $p < 0.05$ and $p < 0.1$, respectively.

conditions, which means that people with good health status tend to participate more in tourism activities. Likewise, income and education have a positive impact on tourist outflows, suggesting that with increased income and education, Chinese people tend to participate more in tourism activities.

The findings suggest that HDI is a significant factor in outbound tourism activities and that improvement in the HDI index can significantly enhance tourist outflows. The component-wise HDI analysis also supports the baseline results.

## 6. Conclusions and implications

In the contemporary world, tourism has emerged as a vital industry that significantly contributes to the economic development of countries. Globalization and increased regional economic integration are the key triggers of tourism activity. Since implementing its reforms and opening-up policy, China has experienced a significant surge in both inbound and outbound tourism. Increased income levels and a rising middle class are the mainstays of a consistently rising demand for outbound tourism by Chinese people [17]. China is now the world's largest contributor to tourism outflows, and recipient countries are competing to further increase the inflow of Chinese tourists [16].

Tourism inflows are an important source of foreign exchange for an economy as they help to create employment opportunities, increase consumption, alleviate poverty, promote economic development, and foster overall human development in a country [19]. Furthermore, heightened tourism activity has led to fruitful interactions between diverse communities and cultures, fostering intercultural harmony and expanding opportunities for economic cooperation among individuals, corporate entities, and countries.

Tourism literature has predominantly focused on the economic aspects of tourism activities, while less attention has been paid to how increased human development stimulates the demand for outbound tourism [6,12]. China has experienced a significant increase in demand for outbound tourism activities over the past few decades. The present research employs ARDL estimation to observe the impact of HDI on China's demand for outbound tourism from 1995 to 2018. GDP and exchange rates are the control variables in our proposed model, as they can significantly influence tourism outflows. Our empirical outcomes confirm a significant influence of improvements in HDI on China's outbound tourism, both in the short and long term. Quite intuitively, GDP and exchange rates have a positive effect on China's tourism outflows, both in the short term and long term.

Furthermore, the influence of individual components of HDI, such as life expectancy, education enrollment, and per capita income, on China's tourism outflow was also explored. A significant positive nexus between longevity and better health conditions, as well as outbound tourism in the long term, suggests that improved health facilities and longevity have contributed to a rise in tourism outflows from China. Higher education enrollments do not show a short-term association with tourism outflows. However, it significantly drives outbound tourism in the long term, reflecting that education creates awareness and inquisitiveness among Chinese people for traveling and exploring new destinations. Nevertheless, per capita income also exhibits a positive association with China's outbound tourism, both in the short term and the long term, implying that the increased disposable income of the Chinese people has substantially boosted demand for leisure and tourism activities.

The outcomes of this study can help academics, researchers, and policymakers understand the impact of overall human development, prosperity, and well-being on the demand for travel and leisure activities within a society. The implications of this research suggest that HDI has a spillover effect on tourism outflows. Hence, small and developing nations should strive to implement favorable policies to attract tourism flows from countries like China, thereby promoting the development of their local economies. Moreover, an increased flow of tourism can also enhance people-to-people exchanges, especially between the Belt and Road countries [75], and foster economic and human development in developing and impoverished economies. Against the backdrop of China's emergence as the world's second-largest economy, tourism outflows can also help China expand its soft power by partnering with Asian countries and beyond.

### 6.1. Limitations of the study

Like other studies, this paper also has some limitations. First, the study is conducted using a single-country dataset, specifically China. Therefore, we cannot generalize these results to other countries, as the legal structure, government, and institutional setting of many nations are significantly different from those in China. Second, studies have predominantly focused on HDI as a single factor influencing tourism outflow, while several other political, economic, social, and psychological factors can also impact outbound tourism.

### Acknowledgments

This research is supported by the Excellence Project at the Faculty of Informatics and Management, University of Hradec Kralove, Czechia.

### Author contributions

**Conceptualization:** Ahsan Akbar.

**Data curation:** Farrukh Nawaz Kayani, Irfan Ullah.

**Formal analysis:** Irfan Ullah.

**Funding acquisition:** Xie Hui, Veronika Zidova, Asokan Vasudevan.

**Methodology:** Irfan Ullah.

**Project administration:** Veronika Zidova.

**Resources:** Farrukh Nawaz Kayani, Xie Hui, Irfan Ullah, Veronika Zidova.

**Software:** Xie Hui.

**Supervision:** Veronika Zidova.

**Validation:** Xie Hui, Minhas Akbar, Veronika Zidova, Asokan Vasudevan.

**Visualization:** Xie Hui, Minhas Akbar, Veronika Zidova, Asokan Vasudevan.

**Writing – original draft:** Ahsan Akbar, Farrukh Nawaz Kayani.

**Writing – review & editing:** Ahsan Akbar, Minhas Akbar, Asokan Vasudevan.

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
