## [Decision Letter · Decision Letter 0]

PONE-D-24-28502Does an Improved HDI Trigger Tourism Outflows for China? New Evidence from ARDL Cointegration ApproachPLOS ONE

Dear Dr. Hui,

Thank you for submitting your manuscript to PLOS ONE. After careful consideration, we feel that it has merit but does not fully meet PLOS ONE’s publication criteria as it currently stands. Therefore, we invite you to submit a revised version of the manuscript that addresses the points raised during the review process. 

We look forward to receiving your revised manuscript.

Kind regards,

Shujaat Naeem Azmi, Ph.D.

Academic Editor

PLOS ONE

Journal Requirements:

3. Thank you for stating the following financial disclosure: “This research is supported by the specific project 6/2024 grant “Determinants of Cognitive Processes Impacting the Work Performance” granted by the University of Hradec Kralove, Czech Republic and thanks to help of student Ing. Frantisek Hasek; a Long-term development plan of the University of Hradec Kralove.”

4. In the online submission form, you indicated that “Data is available with corresponding author and will be provided on reasonable request.”

All PLOS journals now require all data underlying the findings described in their manuscript to be freely available to other researchers, either 1. In a public repository, 2. Within the manuscript itself, or 3. Uploaded as supplementary information. This policy applies to all data except where public deposition would breach compliance with the protocol approved by your research ethics board. If your data cannot be made publicly available for ethical or legal reasons (e.g., public availability would compromise patient privacy), please explain your reasons on resubmission and your exemption request will be escalated for approval.

6. Please include your tables as part of your main manuscript and remove the individual files. Please note that supplementary tables (should remain/ be uploaded) as separate "supporting information" files

Additional Editor Comments:The current state of literature included seems lacking. I encourage authors to explore more literature on tourism to develop a more robust theoretical framework. Particularly how factors like trade influence international trade can influence tourism.  For instance see papers:**1. ** Çalışkan, U., Saltik, I.A., Ceylan, R., Bahar, O.: Panel cointegration analysis of relationship between international trade and tourism: Case of Turkey and silk road countries. Tour. Manag. Perspect. 31, 361–

369 (2019). https:// doi. org/ 10. 1016/j. tmp. 2019. 07. 003**.****2. ** Fischer, C., Gil-Alana, L.A.: The nature of the relationship between international tourism and international

trade: the case of German imports of Spanish wine. Appl. Econ. 41(11), 1345–1359 (2009). https:// doi.

org/ 10. 1080/ 00036 84060 10193 49

3. Gil-Alana, L.A., Fischer, C.: International travelling and trade: Further evidence for the case of Spanish

wine based on fractional vector autoregressive specifications. Appl. Econ. 42(19), 2417–2434 (2010).

https://doi.org/10.1080/00036840701858083.**4. ** Azmi, S. N., Khan, T., Azmi, W., & Azhar, N. (2023). A panel cointegration analysis of linkages between international trade and tourism: case of India and South Asian Association for Regional Cooperation (SAARC) countries. Quality & Quantity, 57(6), 5157-5176.

Reviewers' comments:

Reviewer's Responses to Questions

**Comments to the Author**

1. Is the manuscript technically sound, and do the data support the conclusions?

Reviewer #1: Partly

Reviewer #2: Yes

2. Has the statistical analysis been performed appropriately and rigorously? 

Reviewer #1: Yes

Reviewer #2: Yes

3. Have the authors made all data underlying the findings in their manuscript fully available?

Reviewer #1: Yes

Reviewer #2: Yes

4. Is the manuscript presented in an intelligible fashion and written in standard English?

Reviewer #1: Yes

Reviewer #2: Yes

5. Review Comments to the Author

Reviewer #1: Dear Xie Hui et al.,

It was delightful to read the manuscript PONE-D-24-28502, titled “Does an Improved HDI Trigger Tourism Outflows for China? New Evidence from the ARDL Cointegration Approach.”

Indeed, tourism contributes significantly to people’s and society’s welfare through many channels, but essentially by empowering and providing resources to communities in host countries while expanding tourists' knowledge and experience. The choice of the human development index variable is perfect as it can capture all these benefits and ultimately promote and enhance individuals' mobility.

I particularly like President Deng Xiaoping’s quote, “It does not matter whether the cat is black or white as long as it catches the mice.”

I am very familiar with the model used but not so much with the topic. Some of my research has been on migration and differentiating the economic perspectives of both host and sending communities and countries. Therefore, I was generic to some extent, acknowledging my limitations on the topic by asking questions. Yet, I have also been thorough regarding the scientific expectations from an article on the methods and conclusions.

Below are two sections: a brief summary of my understanding of the study and a comments/question section covering topics from minor to major. I am available to expand on some of my comments if they require further explanation.

Summary:

The study assessed the association between the Human Development Index (HDI) and tourism outflows in China. HDI comprises three main indicators of human development: (i) Life expectancy at birth, (ii) Literacy rate, and (iii) Per capita income of the residents of a country. Outbound tourism has become an essential contributor to countries' sustainable development. It helps to understand cross-cultural similarities and differences while promoting the goodwill of the home country.

To empirically test the relationships linking tourism outflow and HDI, the authors used time series data from 1995-2018. They modeled short and long-run cointegration with a series of five explanatory variables. The short-run and long-run Autoregressive Distributed Lag (ARDL) results suggest that improving Chinese population HDI positively impacts tourism outflows. This implies that a 1% increase in the HDI index will increase tourism outflow by 11.6%. Factors underlying these results are higher education enrollments, life expectancy, and per capita income. Policy recommendations were made based on the study results.

Comments and questions

The reviewer appreciates the literature review supporting the choice of variables and the authors detailing their conceptual framework.

Major comments/questions:

1- However, as they may expect, my first question is whether their data distinguished tourism as the movement of people or mobility for a short visit and migration as the same movement for a more extended period, which does not necessarily include leisure. This is because you repeatedly mentioned education leading to these outflows. Often, countries have different visa types for tourism, i.e., short stays for leisure only and for a few months usually. Education or academic visas separate because of the potential that this education might evolve into "working abroad" or "skill migration." How did you ensure that your data did not overlap some migration movements?

2- It is a no-brainer that per capita income is a deciding factor for both inbound and outbound tourism. So, when using (HDI), Are you somehow insinuating that only educated people are most likely to travel on tourism? Your education variable only mentions enrollment in higher education, not completion of any degree. So, we would need stats on completion and dropout rates to confidently use education as a key deciding factor in the discussion section.

3—Table 1: An exchange rate is a currency-based variable generated compared to another foreign currency, such as USD or EURO to CNY. Unless I missed it, the reviewer has not found any foreign currencies used to estimate the EX variable anywhere in the text. If one country in particular has been used, discussions should be limited to those two countries.

4- GDP (Gross Domestic Product) and PPP are often highly correlated because they both measure economic activities. Since these variables are related, including both in the same model might lead to multicollinearity, making it difficult to identify each variable's effects. This correlation might signal a deeper endogeneity issue, such as reverse causality and omitted variables. Did you test for multicollinearity and endogeneity because the reviewer didn't see any of their results in the text?

5—Line 312: PPP is not a direct proxy for inflation. It is more of a cross-sectional tool for comparing economic well-being between countries. For example, if inflation in one country is significantly higher than in another, the first country's PPP-adjusted GDP might decline relative to the second country.

6- I found some counter-intuitive results in Table 4. How could you explain that when PPP improves, people become less likely to travel on tourism? Meanwhile, the GPD (or GDP PC) suggests otherwise.

7- Line 254 to 257: Hypothesis error

Your hypothesis does not correspond to the general framing of cointegration hypotheses. Your formulation involves testing whether individual coefficients are equal to zero, but that is not the central concern of a cointegration test. Instead, you typically test whether the variables are cointegrated (i.e., whether a long-run relationship exists). Your hypothesis specification does not align with Pesaran, Shin, and Smith's (2001) bounds-testing approach.

The correct hypothesis would be:

• Null Hypothesis (H₀): β1=β2=⋯=β6=0β1=β2=⋯=β6=0 (no cointegration).

• Alternative Hypothesis (H₁): At least one of the βi is non-zero (cointegration exists).

8- You provided GDP per capita in the descriptive statistics table but modeled the actual GDP. The reviewer needs clarification about which of these two variables is actually in the model. Please clarify and stick to it in the text.

9- Could you explain why the population variable is restricted only to 15 to 64 years old? If the idea is that kids (>15) cannot travel by themselves on tourist vacations, then we can also challenge the assumption made for a 15-year-old.

The same goes for the upper-level ages. You mentioned that life expectancy has improved in China, so people now live longer and healthier lives and longevity tends to increase tourist outflows. Still, here you are again, restricting the population age to 64— This is a counterintuitive move. Understanding the ideas behind the 15—and 64-year-old age caps will be helpful.

10—Lines 320 to 323: These model specifications, especially for the first and second models, are alarming for multicollinearity. GDP and income are in the same model as population and life expectancy. Please run some tests and report the results. If no issues are found, the reviewers will be happy with the results. However, it is better to check.

11—Many major cities (countries) or destinations around the world have unique and idiosyncratic effects within the tourism industry, including Paris (France), Rome (Italy), Kenya, Seychelles, Hawaii (USA), and the Maldives... You cannot model tourism outflows from China (the largest outflow contributor) without controlling for these specific effects. Also, the reviewer didn't see any bilateral relationships between China and particular countries. So, I want to ask where the Chinese go the most for tourism. We shouldn't be asking these questions at the end of your paper.

Minor comments/questions

1- The results need to be separated from the discussion. The discussion section needs to situate the results against trends in the current literature. Is there any supporting evidence from other related studies in China and the region?

2- Table 2: ADF unit root test. None of the estimates, even at the difference, have significance levels. Please follow a standard method of presenting these results and add that these are the first differences.

3- Lines 73 to 75: In 1990, the China National Tourism Administration (CNTA) announced “Provisional measures concerning the administration of arranging Chinese citizens to three Southeast Asian countries.” Could you name the three countries mentioned? Lines 123, 148, and 154 provide similar expectations without mentioning countries.

4- Line 84 – 85: “However, the trade deficit reflects that tourism outflows are higher than inflows.” Could you add some statistics and references to this line and expand more than the percentages provided in the subsequent line 86?

5- Line 98: children from the host country have a negative attitude towards tourists (35).” Could you expand on this counterintuitive result and provide some context?

6- Line 126 to 127: What does person-time thousand mean? It is always good practice to explain ratios and units and not assume that the reader will readily perceive them. Also, it will be good to provide some examples of leading destinations in both directions, i.e., inbound and outbound tourist movements.

7- Line 141: “Lim (49) it was…” This sentence looks odd.

8- Line 166: Restructure the sentence ending with North Korea.

9- Lines 243 to 146: If China were among the countries studied by the authors (50), it would be interesting if you could link it to your current case study. At least you could expand on the authors’ results here and include some critical trends.

10- Lines 193 to 198: Are these enrollments into higher education trends in China or abroad universities?

11- Lines 354 - 355: Please list a few countries, like the top three or five nations competing for Chinese tourists, for illustration purposes. This is important because the outflow of tourism is also based on the laws and regulations governing tourism in these host countries.

Reviewer #2: Dear Authors,

Congratulations for the wonderful piece of research work.

The research needs to address following aspects:

1. The background section need to review to make short and the sentence structure must be reviewed. For example, Line 58, 59, 63 64, 79, 107 etc.

2. The 3rd and 4th Paragraph in Background Section looks irrelevant. which must be revisited.

3. The literature review needs more research exploration related to connecting with HDI and its components.

4. The methodology Section must be updated to verify your proposed model. If possible, verify with previous research.

5. The conclusion and implications section is full of evidence from previous studies where the citations are missing.

6. The final paragraph must be separate section: Limitations

Good Luck and wish you all the best.

6. PLOS authors have the option to publish the peer review history of their article (what does this mean? ). If published, this will include your full peer review and any attached files.

**Do you want your identity to be public for this peer review?** For information about this choice, including consent withdrawal, please see our Privacy Policy .

Reviewer #1: No

Reviewer #2: No

---

## [Author Response · Author response to Decision Letter 1]

24 Jun 2025

Please see attached response sheet.

---

## [Editor Report · Decision Letter 1]

Does an Improved HDI Trigger Tourism Outflows for China? New Evidence from the ARDL Cointegration Approach

PONE-D-24-28502R1

Dear Dr. Hui,

We’re pleased to inform you that your manuscript has been judged scientifically suitable for publication and will be formally accepted for publication once it meets all outstanding technical requirements.

Kind regards,

Shujaat Naeem Azmi, Ph.D.

Academic Editor

PLOS ONE

---

## [Editor Report · Acceptance letter]

PONE-D-24-28502R1

PLOS ONE

Dear Dr. Hui,

I'm pleased to inform you that your manuscript has been deemed suitable for publication in PLOS ONE. Congratulations! Your manuscript is now being handed over to our production team.

Kind regards,

on behalf of

Dr. Shujaat Naeem Azmi

Academic Editor

PLOS ONE